# Engineered Cas12i2 is a versatile high-efficiency platform for therapeutic genome editing

Colin McGaw[1,2], Anthony J. Garrity[1,2], Gabrielle Z. Munoz[1], Jeffrey R. Haswell[1], Sejuti Sengupta[1], Elise Keston-Smith[1], Pratyusha Hunnewell[1], Alexa Ornstein[1], Mishti Bose[1], Quinton Wessells [1], Noah Jakimo[1], Paul Yan[1], Huaibin Zhang[1], Lauren E. Alfonse[1], Roy Ziblat[1], Jason M. Carte[1], Wei-Cheng Lu[1], Derek Cerchione[1], Brendan Hilbert[1], Shanmugapriya Sothiselvam[1], Winston X. Yan[1], David R. Cheng[1], David A. Scott[1], Tia DiTommaso [1✉] & Shaorong Chong[1]

The CRISPR-Cas type V-I is a family of Cas12i-containing programmable nuclease systems guided by a short crRNA without requirement for a tracrRNA. Here we present an engineered Type V-I CRISPR system (Cas12i), ABR-001, which utilizes a tracr-less guide RNA. The compact Cas12i effector is capable of self-processing pre-crRNA and cleaving dsDNA targets, which facilitates versatile delivery options and multiplexing, respectively. We apply an unbiased mutational scanning approach to enhance initially low editing activity of Cas12i2. The engineered variant, ABR-001, exhibits broad genome editing capability in human cell lines, primary T cells, and CD34+ hematopoietic stem and progenitor cells, with both robust efficiency and high specificity. In addition, ABR-001 achieves a high level of genome editing when delivered via AAV vector to HEK293T cells. This work establishes ABR-001 as a versatile, specific, and high-performance platform for ex vivo and in vivo gene therapy.

[1] Arbor Biotechnologies, 20 Acorn Park Drive, Tower 500, Cambridge, MA, USA. [2]These authors contributed equally: Colin McGaw, Anthony J. Garrity. ✉email: tditommaso@arbor.bio

C RISPR-Cas (Clustered Regularly Interspaced Short Palindromic Repeats-CRISPR associated proteins) systems are widely distributed in both archaea and bacteria and provide adaptive immunity against invading viruses[1,2]. Class 2 CRISPR-Cas systems, notably type II Cas9 and type V Cpf1 (Cas12a), have been harnessed for therapeutic gene editing[3–6]. To expand the CRISPR therapeutic toolbox, we recently identified a number of functionally diverse type V systems (subtypes V-G, V-H and V-I)[7]. The subtype V-I system (Cas12i) represents an evolutionarily distinct branch that clusters closely with the subtype V-B (Cas12b) (Supplementary Fig. 1a). However, Cas12i functionally resembles Cas12a in the fact that Cas12i can also process precursor crRNA (pre-crRNA) and does not require trans-activating crRNA (tracrRNA) for target DNA cleavage[7,8] (Supplementary Table 1 and references therein). Compared to SpCas9 and Cas12a, Cas12i effectors are smaller (1033 to 1093aa) which, in conjunction with their short mature crRNA (40-43 nt) and other aforementioned features (Supplementary Table 1), makes Cas12is particularly suitable for multiplexed genome editing and viral vector-based delivery. However, in spite of their DNA cleavage activity[7], Cas12is (Cas12i1 and Cas12i2) have not been shown to exhibit therapeutic utility in mammalian cells.

Here we applied a high throughput mutational scanning method to engineer Cas12i2. We identified indel-enhancing single amino acid substitutions, which, when combined, showed robust on-target activity in both immortalized and primary human cells, while retaining the high specificity characteristic of type V CRISPR-Cas systems[9–11]. Furthermore, the engineered Cas12i2 mediated efficient gene disruption in primary T cells and CD34+ hematopoietic stem and progenitor cells (HSPCs) when transfected as a ribonucleoprotein (RNP), and high indel activity in HEK293T cells when delivered via AAV vector, thereby demonstrating potentials for both ex vivo and in vivo therapeutic applications.

## Results

**Wild-type Cas12i2 exhibits low editing efficiency in mammalian cells**. We have previously shown that Cas12i2 exhibited both double-stranded DNA (dsDNA) cleavage and collateral single-stranded DNA (ssDNA) cleavage upon target recognition[7] (Supplementary Fig. 1b, c). To determine whether wild-type Cas12i2 (Cas12i2 WT) can be directly used for genome editing in mammalian cells, we transfected HEK293T cells with plasmids expressing NLS-tagged Cas12i effectors and linear DNA templates expressing guide RNAs (gRNAs) targeting several genomic loci. The editing efficiency was measured by the formation of insertion or deletion (indel) using targeted deep sequencing. Low indel activity was detected across a broad set of genomic loci with a mean indel rate of 1.1% (Fig. 1a and Supplementary Fig. 2). The data indicate that wild-type Cas12i2 does not have sufficient indel activity in mammalian cells for some therapeutic genome editing applications.

**Semi-rational engineering of Cas12i2 to improve indel activity in mammalian cells**. Structure-guided substitutions of amino acid residues with arginine or glycine have been shown to increase editing efficiency of type V systems in mammalian cells[12–14]. We hypothesized that the low editing efficiency of Cas12i2 WT could be overcome by introducing arginine or glycine substitutions at strategic positions that could enhance effector binding to the nucleic acid backbone and promote effector conformational flexibility, respectively. In the absence of a Cas12i structure at the time, we took a semi-rational scanning mutagenesis approach and substituted 480 residues in the

C-terminal RuvC domain-containing nuclease lobe of Cas12i2 with arginine or glycine. To rapidly screen a large number of effector variants, we developed an in vitro assay that couples cell-free protein synthesis with a fluorescent reporter assay[15,16]. This in vitro fluorescent reporter assay measures a decrease (depletion) in green fluorescent protein (GFP) signal as the result of Cas12i2 effector-mediated in vitro cleavage of the DNA template that expresses GFP protein (detailed description in Supplementary Methods and Supplementary Figs. 3–5). We screened a total of 960 arginine and glycine single substitutions (at 480 positions) of Cas12i2 in the in vitro fluorescent reporter assay. Based on the GFP signal depletion relative to the wild-type controls, we identified a number of variants showing a significant increase in GFP depletion (with a Z-score>2) for at least one GFP target (Supplementary Fig. 6a, b). A majority of arginine and glycine substitutions, however, did not enhance the GFP signal depletion activity (Supplementary Fig. 6a, Z score < =1). 14 of these in vitro hits were tested for indel activity at two genomic target sites in HEK293T cells (Fig. 1b). Three substitutions, D581R, I926R and V1030G, showed 1.5- to 2-fold improvement in indel activity relative to Cas12i2 WT at one or both target sites (Fig. 1b). The rest of the in vitro hits showed minimal improvement, or even decreased indel activity (Fig. 1b), suggesting underlying differences between cell-free in vitro activity and cellular editing function. Combining the top three substitutions resulted in a variant with at least 3-fold improvement in indel activity relative to Cas12i2 WT, suggesting that the effect of these single mutations was additive (Fig. 1c). This combination variant was chosen as Cas12i2 v1 and named ABR-001.

To test ABR-001 activity, we designed an experiment targeting a broad set of 18 genomic target loci in HEK293T cells. In this experiment, ABR-001 exhibited indel activity approaching SpCas9 efficiency (Fig. 1d). Notably, analysis of indel patterns formed by ABR-001 targeting these genomic loci in HEK293T cells identified prominent large deletions of 5–20 nucleotides (nts), in contrast to small deletions and +1 insertion observed with SpCas9[17] (Fig. 1e). Consistent with these data, ABR-001 showed increased dsDNA cleavage activity relative to Cas12i2 WT in in vitro biochemical assays (Supplementary Fig. 7). Deep sequencing analyses of DNA fragments from Cas12i2 WT in vitro cleavage reactions identified a number of cut sites in the PAM-distal DNA region on both the template strand (TS) and non-template strand (NTS), similar to those observed in Cas12a[18], but with a wide distribution of NTS cut sites outside of the R-loop (Supplementary Figs. 8–11). The extended distribution of cut sites does not seem to be caused by the substitutions in ABR-001 as the downstream cut sites were also found with Cas12i2 WT (Supplementary Fig. 11).

**ABR-001 is a specific nuclease for genome editing**. To comprehensively assess ABR-001 specificity, we employed an unbiased approach of tagmentation-based tag integration site sequencing (TTISS)[19] which, similarly to GUIDE-seq[20], uses the integration of a known donor oligo to detect dsDNA breaks in an unbiased manner. TTISS allowed us to generate an empirical list of potential off-target sites for 18 targets across three genomic loci for both ABR-001 and SpCas9. For instance, at *VEGFA* target 1, no off-target was detected for ABR-001, whereas one major and six minor off-target sites were identified for SpCas9 (Fig. 2a). Aggregating all 18 targets, TTISS identified more potential off-target sites with a higher number of unique integration events for SpCas9 than ABR-001(Fig. 2b–d, Supplementary Fig. 12). Alternatively, we took a prediction-based approach[21]. To identify most likely off-target sequences via in silico prediction, the human genome

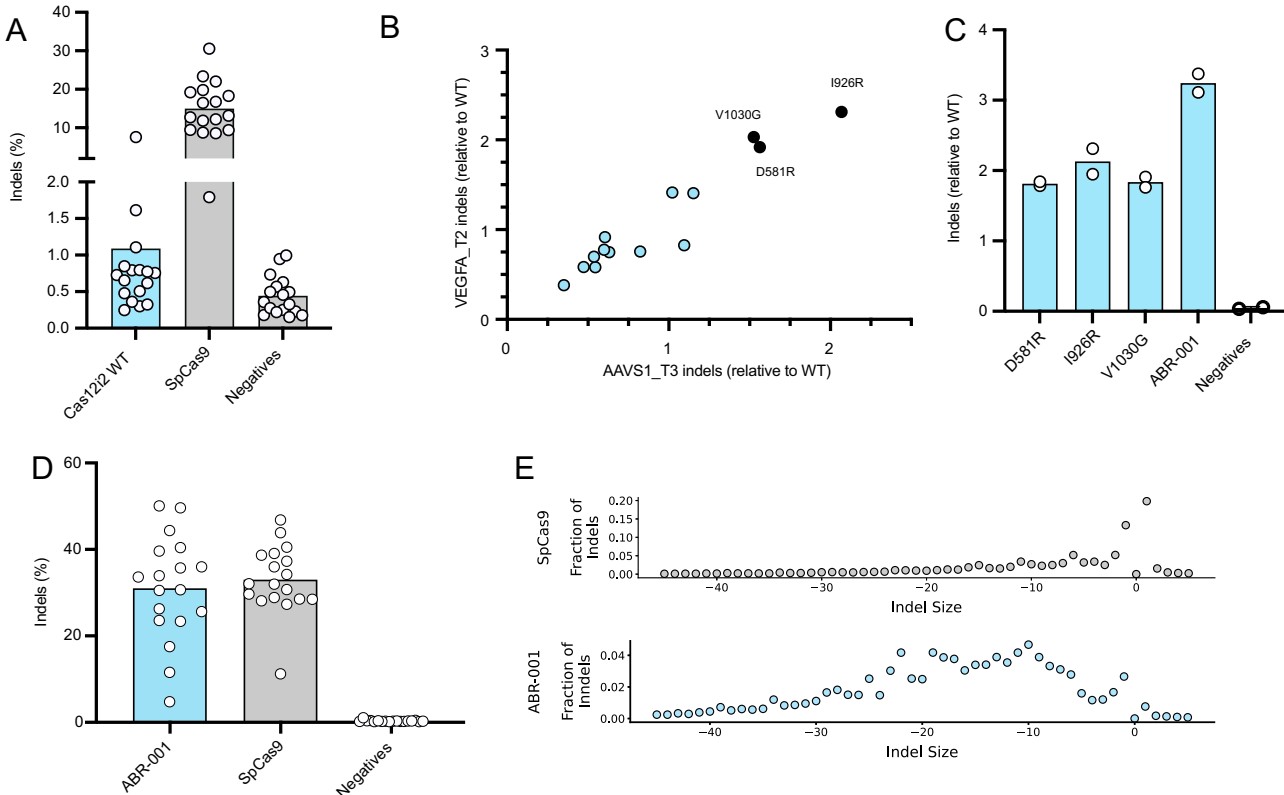

**Fig. 1 Semi-rational engineering of Cas12i2 to improve indel activity in mammalian cells. a** Indel activity of wild-type Cas12i2 (WT) and SpCas9 expressed from a EFS promoter in HEK293T cells at 18 genomic target sites. Each circle represents a single target site, averaged from $n = 2$ replicates. Each bar represents the mean across 18 targets (ABR-001 in blue, SpCas9 and negative in gray). **b** Indel activity (relative to WT) of 14 single-substitution variants measured in HEK293T cells at two genomic target sites. Each circle represents indels for a single variant ($n = 1$). Black circles indicate the three mutations present in ABR-001. **c** Indel activity (relative to WT) of top three single substitutions and their combination variant, ABR-001. Bars represent mean of 2 targets and each circle represents $n = 2$ replicates at a single target site. **d** Indel activity of ABR-001 and SpCas9 expressed from a CMV promoter in HEK293T cells at 18 genomic target sites. Each circle represents a single target site, averaged from $n = 2$ replicates. Each bar represents the mean across 18 targets (ABR-001 in blue, SpCas9 and negative in gray). e. Indel size distribution comparison between SpCas9 and ABR-001 after genome editing of 18 target sites in HEK293T cells. The fraction of indels is analyzed by aggregating all indel reads, binning the reads by indel size (a minus value is deletion, a positive value is insertion). Circles represent average fraction of indels, binned by indel size, of 18 targets with $n = 2$ replicates (ABR-001 in blue, SpCas9 in gray). Source Data are provided as a Source Data file.

was searched for sequences adjacent to the PAM for each of the 18 targets in the expanded set. Off-targets were predicted and ranked by ascending edit distance—the number of insertions, mismatches, and deletions between the on-target and off-target sequence. The editing efficiencies of the top 10 off-target sites were characterized using deep sequencing and background noise was corrected using maximum likelihood estimation (MLE). Using this approach, off-target indels greater than the limit of detection (>0.2%) were identified in 2 out of 18 targets for ABR-001 compared to 8 out of 18 targets for SpCas9 (Supplementary Fig. 13). However, the sparsity of identified off-targets could be due to limitations in the number of off-targets screened with this biased prediction-based approach. Taken together, these data demonstrate potential therapeutic utility of ABR-001 as an active and specific genome editing nuclease.

**ABR-001 ribonucleoproteins (RNPs) mediate ex vivo editing of human primary T cells and CD34+ HSPCs.** To test the therapeutic potential of ABR-001 for ex vivo cell therapy, we generated ABR-001-gRNA complexes and delivered the ribonucleoproteins (RNPs) into stimulated human CD3+ T cells by electroporation at various concentrations. We first targeted

β2 microglobulin (*B2M*) with 4 different gRNAs and analyzed cells at 7 days post-electroporation for editing efficiency, cellular viability and B2M protein expression. All four B2M guides resulted in robust editing with positively correlated dose response (Fig. 3a), >70% viability for all concentrations tested (Fig. 3b), and efficient B2M protein knockdown (Fig. 3c and Supplementary Fig. 14). The optimal dose was determined to be 16μM, a concentration at which indel formation was close to 90% and viability was maintained at >70%. We next targeted two additional therapeutically relevant genes, *TRAC* and *CIITA*. Again, robust indel efficiency (Fig. 3d) and target knockdown (Fig. 3f) were observed along with maintenance of cellular viability (Fig. 3e). These data indicate that ABR-001 RNPs can be used for editing therapeutically relevant targets in human T cells at RNP concentrations that have no impact on cell viability.

To further characterize the indel pattern of ABR-001 editing in primary cells, we assessed the fraction of indels at a given indel size using data from the T cell experiments. Again, ABR-001 editing was biased toward large deletions in primary cells (Fig. 3g), similar to our initial observations in HEK293T cells (Fig. 1e). Further, the deletion length (indel size) and relative frequency appeared to be target dependent. Specifically, the most frequent deletions were >19 nt for *TRAC*, and 10-17 nt for

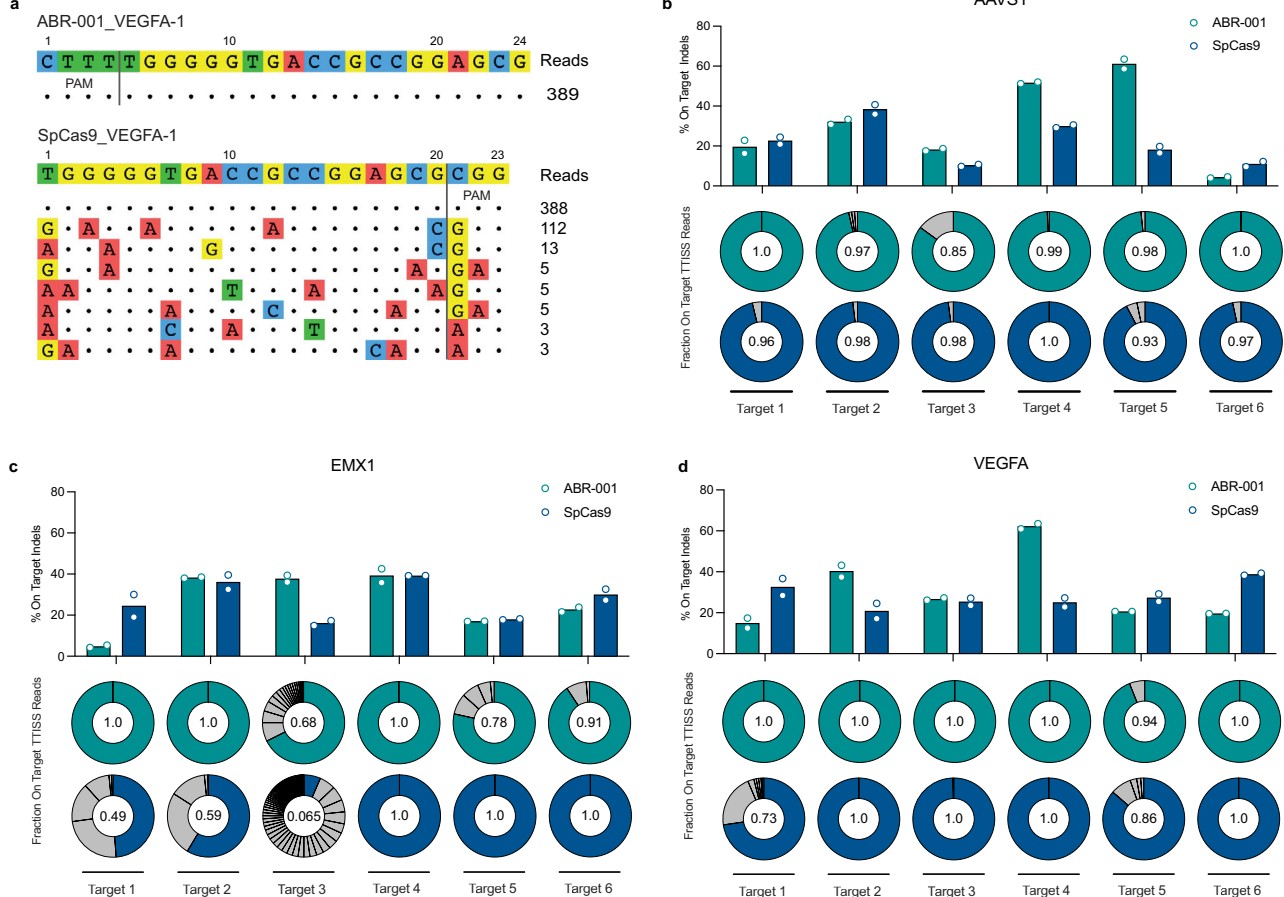

**Fig. 2 ABR-001 is a specific nuclease for genome editing. a** Representative off-target discovery of ABR-001 and SpCas9 by TTISS at VEGFA Target 1.
**b–d** TTISS analysis across 6 test loci for each of 3 genes. Teal (ABR-001) and blue (SpCas9) bars show average on-target indel activity for $n = 2$ replicates,
each indicated by an outlined circle. Doughnut plots depict proportion of detected cleavage sites for each nuclease. Off-targets are shown as light gray
wedges, while the on-target site is highlighted in teal (ABR-001) or blue (SpCas9). Centered numerical values represent fraction of on-target TTISS reads.
Source Data are provided as a Source Data file.

*CIITA* (Fig. 3g). While deletion lengths vary between targets, deletion patterns were highly conserved across independent experiments (Fig. 3g), suggesting the editing was a non-random process.

To test the therapeutic potential of ABR-001 for ex vivo cell therapy, we delivered ABR-001 RNPs targeting the erythroid specific *BCL11A* enhancer in CD34+ hematopoietic stem and progenitor cells (HSPCs). Three guides were designed to generate indels within the *BCL11A* enhancer region to disrupt the GATAA motif and induce fetal hemoglobin (HbF) in adult CD34+ HSPCs[22,23]. All three guides generated robust indel rates comparable to SpCas9 within the enhancer region, two of which showed significant disruption of the GATAA motif, albeit at lower levels than SpCas9 (Fig. 3h). The dual guide combination (multiplex target 1+2) maximized the GATAA motif disruption as well as overall indel rates (Fig. 3h). Therefore, we selected the single guide (target 2) and dual guides (multiplex target 1+2) for further functional studies.

We used in vitro colony-forming cell (CFC) assays to test the function of ABR-001 edited within the *BCL11a* enhancer region in HSPCs and showed that editing with ABR-001 in vitro did not interfere with HSPC multilineage differentiation of these edited cells. Counts for the erythroid, myeloid, and mixed colonies were comparable among untreated samples and samples electroporated with ABR-001 protein

or ABR-001 RNPs (Supplementary Fig. 15a). In the conditions containing ABR-001 RNPs, the indel rates detected in isolated colonies at day 15 matched or exceeded indel rates of the mixed input population at 72 h after electroporation suggesting that progenitor cells were successfully transfected in this in vitro assay (Supplementary Fig. 15b)

To evaluate indel persistence and HbF induction, CD34+ cells electroporated with ABR-001 and SpCas9 RNPs were expanded in erythroid differentiation medium[24]. Cells were sampled at 3 and 20 days post-electroporation and analyzed for indel persistence and HbF expression. High levels of editing by ABR-001 were maintained throughout the 20-day erythroid differentiation period for both single and dual guides (Fig. 3i), demonstrating that ABR-001 edited cells can persist long-term in vitro. The characteristic broad deletion profile of ABR-001 was retained in the HSPCs post differentiation (Fig. 3j). Further, the ABR-001 multiplex target 1+2 enabled a broader deletion profile, as compared to the single target 2 (Fig. 3j). HbF expression was assessed post 20-day erythroid differentiation period, showing a high level of fetal hemoglobin induction in the ABR-001 edited cells (Fig. 3k). Interestingly, ABR-001 edited cells using both single and dual guides yielded robust and SpCas9-equivalent HbF expression (Fig. 3k), despite lower levels of indel rates and GATAA disruption relative to SpCas9 (Fig. 3i). These results

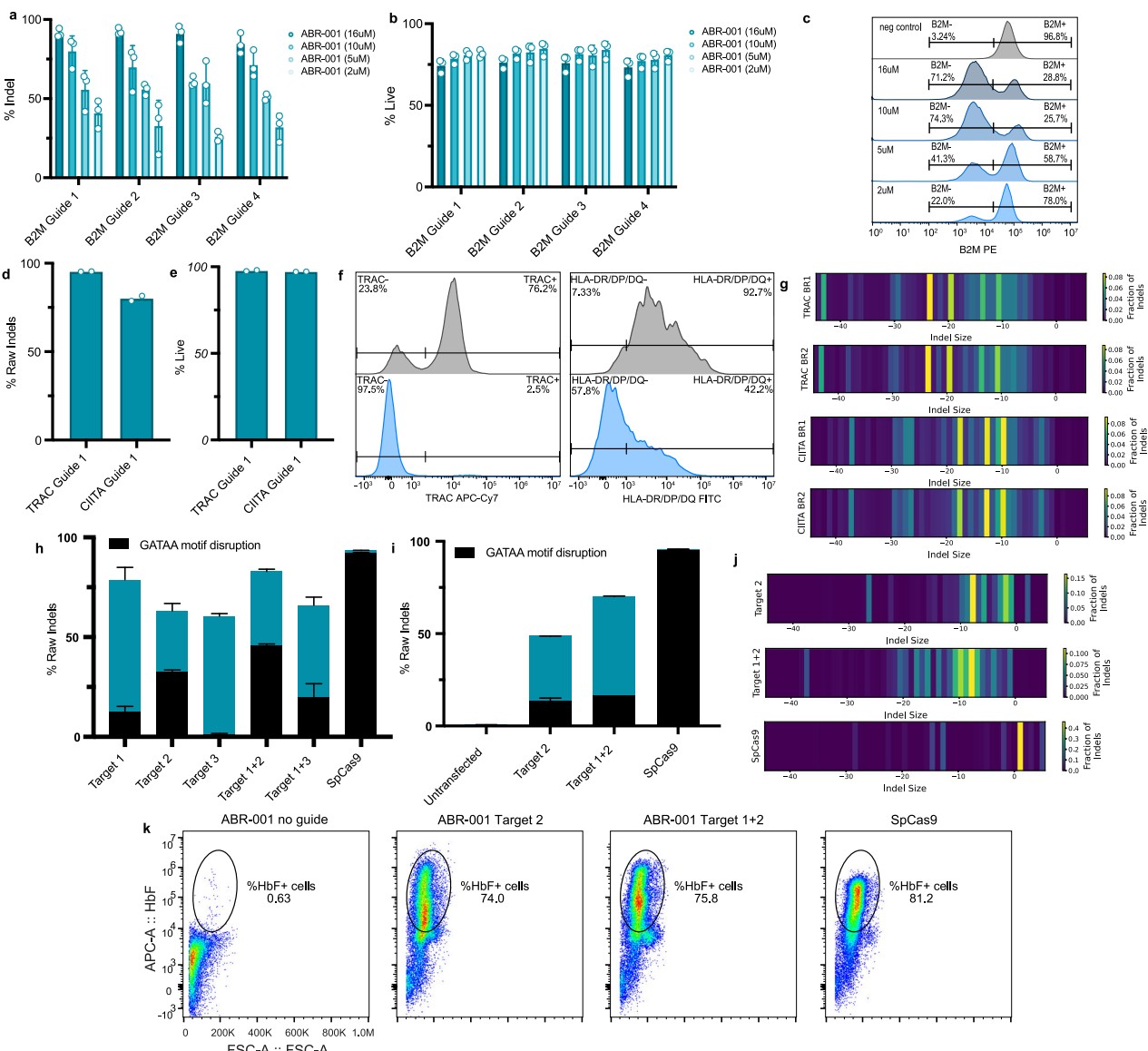

**Fig. 3 ABR-001 enables ex vivo editing of human primary cells. a**, **b** Indel activity (**a**) and viability (**b**) of human CD3 + T cells following delivery of ABR-001 RNPs with 4 different B2M-targeting guides (B2M 1–4) at decreasing concentrations (16 µM, 10 µM, 5 µM, and 2 µM). Cell viability was measured using flow cytometry at 7 days post-electroporation. Bars represent the average of 3 donors (circles), with error bars representing the s.d.. **c** Representative histograms show B2M protein level knockdown at 7 days post-electroporation with ABR-001 RNP with B2M 1 guide at decreasing concentrations (16 µM, 10 µM, 5 µM, and 2 µM). Y-axes represent count normalized to mode. **d** Indel activity of CD3+ T cells following delivery of 16 µM ABR-001 RNPs targeting TRAC and CIITA. Circles represent each of $n = 2$ bioreplicates. **e** Viability of CD3+ T cells following delivery of 16 µM ABR-001 RNPs targeting TRAC and CIITA. Circles represent each of $n = 2$ bioreplicates. **f** Representative histograms show protein level knockdown at 7 days post-electroporation for TRAC and CIITA (HLA-DR/DP/DQ readout). Y-axes represent count normalized to mode. **g** Indel size was compared between two independent experiments (BR1 and BR2) of ABR-001 RNPs targeting TRAC and CIITA. The color scale indicates the fraction of indel reads at a given indel size. **h** Indel rates (teal) and % GATAA disruption (black) of multiplexed ABR-001 RNPs (16 µM total concentration) or single SpCas9 RNP (5 µM concentration) targeting the BCL11A enhancer, 72 h post-delivery into human CD34+ cells. Bars represent mean and error bars represent s.d. of $n = 3$ replicates. **i** Indel rates (teal) and % GATAA disruption (black) of ABR-001 RNPs (20 µM total concentration) and SpCas9 RNP (5 µM concentration) after delivery and 20 days of maturation in erythroid differentiation media. Bars represent mean and error bars represent s.d. of $n = 3$ replicates. **j** Indel profiles of ABR-001 RNP (target 2 and multiplex target1+2) and SpCas9 RNP targeting the BCL11A enhancer in human CD34+ cells. The color scale indicates the fraction of indel reads at a given indel size. **k** Representative flow cytometric analyses of edited CD34+ cells after delivery and 20 days of maturation in erythroid differentiation media. Source Data are provided as a Source Data file.

suggest that disruption of the GATAA motif and a broad region of the enhancer flanking the GATAA motif may be beneficial for inducing HbF expression, and by extension, that the broad indel profile could uniquely position ABR-001 for disrupting non-coding genetic elements to achieve positive therapeutic outcomes.

These in vitro results prompted us to test the persistence of ABR-001 edited cells in vivo. CD34+ HSPCs from a single donor were electroporated with ABR-001 and SpCas9 RNPs. Three days following electroporation, edited cells were transplanted into irradiated immunodeficient NSG mice at a dose of 200,000 cells. Indel analyses at the time of adoptive cell

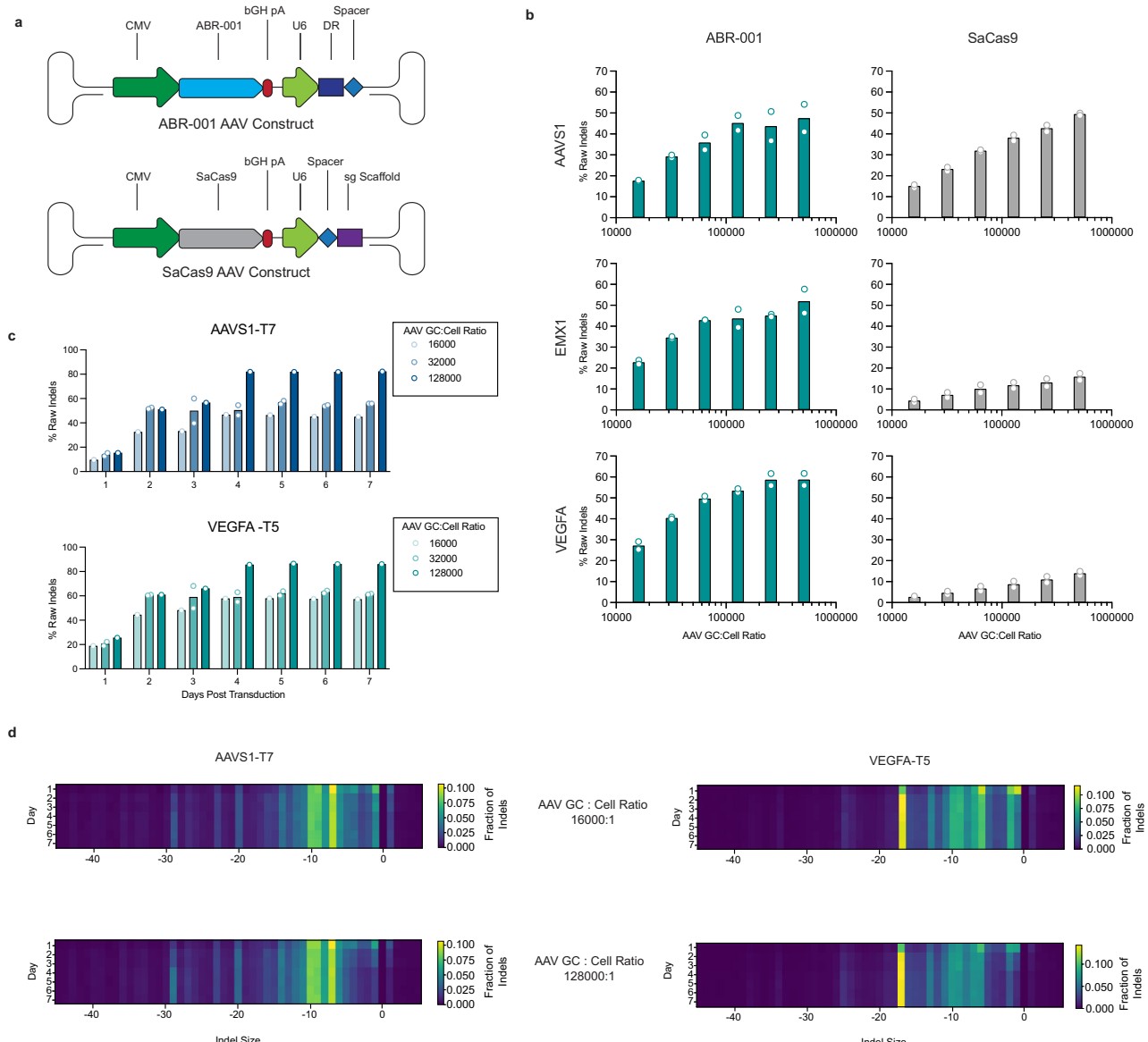

**Fig. 4 ABR-001 mediates highly efficient genome editing when delivered via AAV vector. a** Schematic of the AAV vector backbone for delivery of ABR-001 effector and a guide crRNA. **b** Indel rates at 72 h following transduction of ABR-001 and SaCas9 AAVs at increasing MOI in HEK293T cells. Teal (ABR-001) and gray (SaCas9) shaded bars represent the average of *n* = 2 bioreplicates, each indicated by an outlined circle. **c** Indel rates at two target sites measured daily for 7 days after transduction of HEK293T cells with ABR-001 AAVs at GC:Cell ratios of 1.6e4, 3.2e4, and 1.3e5. Shaded bars represent average indel rate of *n* = 2 bioreplicates for GC:Cell ratio of 3.2e4 and *n* = 1 bioreplicate for GC:Cell ratio of 1.6e4 and 1.3e5. Measurements for individual bioreplicates are indicated by an outlined circle. **d** ABR-001-AAV indel profiles across two targets and 2 MOIs over a time course of 7 days. The color scale indicates the fraction of indel reads at a given indel size. Source Data are provided as a Source Data file.

transfer and at 8 and 16 weeks post-engraftment showed persistent levels of editing over time (Supplementary Fig. 16a). ABR-001 edited human CD45+ cells harvested from the bone marrow showed SpCas9-equivalent HbF expression, although a significant HbF induction relative to the untransfected group was not observed (Supplementary Fig. 16b). Importantly, ABR-001 edited human cells showed similar and, in some cases, higher in vivo engraftment capacity than unedited or SpCas9 edited cells at 8 or 16 weeks post-transplantation (Supplementary Fig. 13c–f). Taken together, these data demonstrate potent long-term persistence and functionality of ABR-001-edited CD34+ HSPCs.

**ABR-001 achieves highly efficient genome editing when delivered via AAV vector.** With its compact size and lack of

tracrRNA, ABR-001 is an ideal candidate for AAV delivery. There have been few demonstrations of efficient editing with type II systems when delivered via AAV vector[25,26]. To enable the AAV delivery, we constructed an AAV2 vector expressing the ABR-001 effector under a CMV promoter and gRNA under a U6 promoter (Fig. 4a). Indel levels were measured at 72 h following transduction of HEK293T cells at varying multiplicities of infection (MOI). We found increasing editing levels that correlated with increasing MOI. With the highest MOI, the indel rate reached up to 60%, equivalent to or exceeding the levels for SaCas9 at matched target sites (Fig. 4b). To test the editing persistence, cells were cultured for up to 7 days post-transduction. We found that cells maintained high indel levels after day 1, increasing up to >80% by day 4 (Fig. 4c) and indel patterns remained relatively

consistent after day 4 (Fig. 4d). The jump in indel activity seen between day 3 and day 4 is likely due to the expansion of cells that occurred on day 3. As observed with non-viral delivery methods, the indel pattern was target dependent (Fig. 4d). Under these conditions, our data show that ABR-001 can achieve high editing levels in vitro when delivered by AAV and therefore suggest that ABR-001 may be a promising candidate for development into an in vivo gene editor.

## Discussion

Rapid growth in genomic and metagenomic sequence databases has led to discovery of an increasing number of CRISPR-Cas systems, including many functionally diverse type V-I nuclease systems[7,24]. All CRISPR-Cas systems originate from bacteria, archaea, or bacteriophage[24], and therefore, are not evolved for efficient genome editing in mammalian cells. Prior to our work, Cas12i has not been utilized for mammalian genome editing. Here we demonstrate that by introducing arginine and glycine mutations at critical positions of the C-terminal RuvC domain-containing nuclease lobe, we can improve Cas12i2 editing efficiency to achieve comparable editing outcomes to those published for edited cell therapies[27,28]. Our results with ABR-001, along with data from other studies[12,13,29], suggest that at least one critical factor limiting therapeutic applications of Cas12i and perhaps other type V systems might be the weak interaction of a CRISPR-Cas effector or RNP with mammalian genomic target DNA. In agreement with this hypothesis, ABR-001 exhibited higher binding affinity on dsDNA target compared to Cas12i2 WT (Supplementary Fig. 17), suggesting that the combined substitutions of D581R, I926R and V1030G in ABR-001 enhanced the catalytic activity of the effector through tighter binding of DNA substrates. In addition, modeling of D581R and I926R in Cas12i structure[30,31] publications subsequent to the generation of ABR-001indicates that these WED and Nuc domain substitutions could strengthen the electrostatic interaction with the dsDNA backbone of the PAM region and non-template strand DNA, respectively (Supplementary Fig. 18). The residue V1030 is located in the RuvC domain which functions to bind and cleave DNA substrate. The exact mechanism of the V1030G substitution in enhancing indel activity is not clear due to the lack of local structural information. It is possible that the V1030G substitution might influence target DNA interaction in a way that favors DNA cleavage. Rational approaches to engineering using recently published Cas12i structural information[30,31] may further improve Cas12i2 editing efficiency and targeting specificity[12,13,32,33].

Cas12i is an attractive type V CRISPR-Cas nuclease for genome editing because of its compact size that fits in AAV vector with short 43-mer gRNA, absence of tracrRNA, ability to process pre-crRNA, and high specificity. With perhaps the exception of the recently discovered type V-J Cas12j[34,35], no other CRISPR-Cas nuclease possesses this unique combination of favorable properties (Supplementary Table 1). The robust activity and high specificity shown by the first engineered version, ABR-001, establishes Cas12i2 as a versatile genome editing platform with broad capabilities. Notably, ABR-001 demonstrated overall higher indel activity relative to the SaCas9 control with AAV delivery in HEK293T cells. Taken together, ABR-001 represents a high-performance genetic medicine platform with potential for both ex vivo and in vivo gene therapy.

## Methods

**Generation of DNA constructs for in vitro and mammalian cell screens and gene editing applications**. For cell-free protein synthesis and the in vitro fluorescence reporter assay, the genes for SpCas9 and Cas12i2 WT effectors were cloned into pET28 vector (EMD Milipore #69864) and expressed under the T7 promoter. The genes for the GFP target and the RFP control were cloned into a pUC19 plasmid (New England Biolabs #N3041S) and expressed from a $\sigma^{28}$ promoter (fliC). A linear DNA template for each effector variant was generated by PCR using a flanking primer pair to amplify the region of plasmid DNA that contains the open reading frame (ORF) of the effector, upstream T7 promoter and downstream T7 terminator. After PCR, the linear DNA templates (4–5 kb) were purified using CleanNGS SPRI beads (Bulldog Bio #CNGS500). Linear templates for 960 arginine and glycine single substitutions (Cas12i2 aa residues 575-1054) were generated by an overlapping PCR approach with mutagenic oligos, and their protein expression in the reconstituted cell-free protein synthesis system was visualized via gel electrophoresis (Supplementary Methods and Supplementary Fig. 19a–c). For arginine or glycine residues present in Cas12i2 WT, the same overlapping PCR approach was used with non-mutagenic oligos. Linear DNA templates for expressing gRNAs were generated by PCR to include the T7 promoter and an artificial 3' hairpin (hp) sequence that helps protect RNAs from degradation[36]. GFP-targeting gRNAs contain spacer sequences complementary to sequences within the GFP gene, whereas a single non-target gRNA contains a spacer sequence complementary to a region of the pUC19 plasmid outside the GFP gene. Linear DNA template for the $\sigma^{28}$ subunit was generated by PCR and included an upstream T7 promoter and downstream T7 terminator.

For gRNA expression in transient transfection experiments in Fig. 1a, b, linear gRNA constructs were generated to express gRNAs from a U6 promoter as PCR amplicons[8]. For gRNA expression in transient transfection experiments in Fig. 1c–e, pUC19-derived plasmids were constructed to express gRNAs from a U6 promoter. For effector expression in transient transfection experiments in Fig. 1a–c, Cas12i2 WT, single-substitution variants, ABR-001, and SpCas9 were cloned into pcDNA3.1(+) vector (Genscript) to be expressed from an EFS promoter. For the effector expression in transient transfection experiments in Fig. 1d, e, ABR-001, and SpCas9 were expressed from a CMV promoter. All plasmids used for transient transfections were prepared at Genewiz using Maxi- or Mega-prep with an endotoxin-free wash.

For AAV vectors, the constructs for SaCas9 control were generated by cloning SaCas9 guides into a commercially available AAV2-SaCas9 vector[25] (Vector Biolabs). A ABR-001 parent AAV construct was produced by subcloning ABR-001 into the AAV2-SaCas9 vector and simultaneous removal of the SaCas9 sg scaffold. ABR-001 crRNAs containing Cas12i2 DR and spacer were produced by Golden-Gate reaction in which complementary oligos coding for the DR and spacer were annealed for directional ligation into an expression vector[37]. All AAV2 plasmids were prepared by Maxi- or Mega-prep with an endotoxin-free wash. AAV2 was produced at Vector Biolabs using a helper-free triple transfection method. HEK293T cells were transfected with the given effector transfer plasmid plus packaging plasmids. AAV was extracted by a triple freeze/thaw and purified by ultracentrifugation with a CsCl gradient. Viral titer was performed by qPCR using a probe specific for the bGH poly-A signal. Virus was resuspended in PBS plus BSA.

**In vitro fluorescence reporter assay for high throughput screening of Cas12i2 single-substitution variants**. An in vitro fluorescence reporter assay was designed to couple cell-free protein synthesis of a CRISPR-Cas system with a fluorescence reporter activity (detailed description in Supplementary Methods). The reconstituted cell-free protein synthesis system[38] was chosen for its designability and minimal level of nuclease contamination. The reconstituted cell-free system reagents containing the E. coli transcriptional and translational machinery were made according to protocols from previous works[15,39]. Briefly, the cell-free system (manufactured at Arbor Biotechnologies) was supplemented with E. coli RNA polymerase core enzyme (1 U/µL) and murine RNase inhibitor (40 U/µL) (New England Biolabs #M0550S and #M0314S). To set up each 1.2 µL reaction in a 384-well plate, a MANTIS Microfluidic Liquid Handler (FORMULATRIX) was used to dispense a master mix (1 µL) containing the cell-free protein synthesis reagent, 0.04 units E. coli RNA polymerase core enzyme, 1.6 units murine RNase inhibitor, 2 ng GFP plasmid DNA, and 1 ng RFP plasmid DNA to a well prefilled with 10 µL mineral oil. Then an Echo 650 liquid handler (Beckman Coulter) was used to dispense 4 ng linear DNA template of SpCas9, Cas12i2 WT, or Cas12i2 variant effector (40 nL), 0.2 ng linear DNA templates for target or non-target gRNAs (80 nL), and 0.1 ng linear DNA template of the $\sigma^{28}$ subunit (40 nL) to each well. After being sealed with an adhesive film, the 384-well plate was centrifuged to mix all components at the bottom of the well with mineral oil covering on top to prevent vaporization. The reaction was initiated when the plate was placed in a microplate reader (TECAN) and incubated at 37 °C. The kinetic GFP and RFP fluorescence was measured every 10 min for 12 hr. The endpoint fluorescence was measured after incubation at 37 °C for 12 h.

Cas12i2 single amino acid substitution variants were screened in $n = 3$ replicates in 384-well plates. Each plate also contained several arginine-to-arginine and glycine-to-glycine substitutions as wild-type controls. To determine the depletion value, both GFP and RFP endpoint fluorescence was measured for reactions with non-target and target gRNA. After normalizing GFP fluorescence by RFP fluorescence, the ratio of fluorescence in non-target gRNA reactions over fluorescence of target gRNA reactions was calculated as the depletion value. The average depletion values of single-substitution variants and wild-type controls from $n = 3$ replicates were used for Z-score analysis. Each variant was normalized

according to the mean and standard deviation of the wild-type controls used within the same plate. Let $r_{v,t,p}$ be the depletion ratio measured for variant $v$, technical replicate $t$, and plate $p$. For a given plate $p$, set of wild-type variants $W_p$, and set of technical replicates $T_p$, let the plate control mean and standard deviation be given by

$$\mu_p = \frac{\sum\limits_{w \in W_p} \sum\limits_{t \in T_p} r_{w,t,p}}{|W_p||T_p|} \tag{1}$$

$$\sigma_p = \sqrt{\frac{\sum_{w \in W_p} \sum_{t \in T_p} (r_{w,t,p} - \mu_p)^2}{|W_p||T_p|}} \tag{2}$$

The wild-type control-normalized $Z$ score for a variant $v$ screened in plate $p$ was computed as

$$Z_v = \frac{\left(\sum\limits_{t \in T_p} \frac{r_{v,t,p}}{|T_p|}\right) - \mu_p}{\frac{\sigma_p}{\sqrt{|T_p|}}} \tag{3}$$

Only the data of variants with a coefficient of variation (CV) less than 20% were used; Those data of variants with CV > 20% were discarded. Furthermore, in cases where GFP or RFP fluorescence did not reach a minimum value of 60 rfu those data were discarded.

**Purification of ABR-001 protein.** The genes for Cas12i2 WT and ABR-001 were cloned in a pET28-derived expression vector that introduced an N-terminal 6xhis tag and a C-terminal nuclear localization signal (NLS) tag. The plasmids were transformed into *E. coli* BL21 (DE3) (MilliporeSigma #70236-3) cells for protein expression. Cells were grown at 37 °C in Terrific Broth in 2.5 L Thomson Ultra Yield flasks (Thomson #931136-B) until OD600 reached ~0.7–0.8. The temperature was then lowered to 20 °C. Protein expression was induced with 0.5 mM IPTG for 16–20 h before harvesting and freezing cells at −80 °C. Cell paste was suspended in a lysis buffer (25 mM HEPES, pH 7.5, 500 mM NaCl, 5% glycerol, and 0.5 mM TCEP (ThermoFisher #77720). Cells were lysed using a cell disruptor (Constant Systems CF1). Polyethylenimine (PEI) (MilliporeSigma #408727) (5%) was slowly added to the cleared lysate to a final concentration of 0.2% to precipitate nucleic acids. After centrifugation to remove the precipitates, the supernatant was loaded on a Sulfate 650F column equilibrated with Sulfate A buffer (20 mM Bis–Tris, pH 6.5, 5% glycerol, and 0.5 mM TCEP) and the bound proteins are eluted with 100-2000 mM NaCl gradient. The pooled peak fractions were diluted to ~250 mM NaCl and loaded onto a Q Sepharose FF column equilibrated with Q buffer (25 mM Tris-HCl, pH 8.0, 250 mM NaCl, and 0.5 mM TCEP). The flow-through fractions were pooled and concentrated to a final concentration of 400 μM. Fractions were visualized via gel electrophoresis and the purified proteins were stored at −80 °C (Supplementary Fig. 20).

**In vitro RNA synthesis.** All mature crRNAs were purchased from IDT or generated by in vitro transcription of dsDNA PCR templates. All pre-crRNAs were generated by in vitro transcription of dsDNA PCR templates. In vitro transcription was performed using the HiScribe T7 High Yield RNA synthesis kit (New England Biolabs #E2040S) at 37 °C for 3 h followed by treatment with Turbo DNase (Thermo Fisher #AM2238). The RNA was purified using Zymo RNA clean and concentrator (Zymo Research #R1013) using the manufacturer's instructions.

**In vitro cleavage reactions and electrophoretic mobility shift analysis (EMSA).** For in vitro cleavage analyses, DNA substrates were generated by PCR amplification using IR800 and IR700 labeled forward and reverse primers, respectively, resulting in dsDNA targets with IR800 labeled template strand and IR700 labeled non-template strand. The PCR products were cleaned up using CleanNGS SPRI beads at a 1.8x ratio of beads-to-PCR product. 1 μM purified Cas12i2 effectors were pre-incubated with 2 μM crRNA to form RNP in 1x NEBuffer 3 (New England Biolabd #B7003S) at 37 °C for 10 min.NEB B7003S) at 37 °C for 10 min. In vitro cleavage reactions containing 100 nM dsDNA substrates mixed with serial diluted RNP (7.8 nM-1 μM) in 1x NEBuffer 3 were performed at 37 °C for 1 hr. The reactions were quenched with 50 mM EDTA. The reactions underwent RNase cocktail treatment (37 °C for 15 min), followed by Proteinase K treatment (37 °C for 15 min). The reactions were analyzed by denaturing gel electrophoresis using 15% TBE-Urea gels and imaged on an Odyssey CLx (LI-COR) imager.

For in vitro cleavage fragment analysis to determine the cut positions, the reaction products were purified using SPRI beads and Isopropyl alcohol (IPA SPRI). The purified reaction was split into two halves. One half was treated with mung bean nuclease (New England Biolabs #M0250S) at 30 °C for 30 min to remove all 5′ and 3′ overhangs to generate blunt ends, followed by purification with IPA SPRI. Both the mung bean nuclease-treated and -untreated halves were then prepared for sequencing using NEBNext Ultra-II DNA library prep kit (New England Biolabs #E7645S) using manufacturer's instructions. Semi-targeted amplification was used to amplify 5′ and 3′ cut products separately for each sample. All amplicons were pooled and gel extracted prior to sequencing. For each sample,

5′ and 3′ cut product results for mung bean nuclease treated and untreated samples were combined to obtain the full cleavage pattern. The 5′ and 3′ cut product on non-treated samples indicated the cut sites on the non-target strand and the target strand respectively, while the 5′ and 3′ cut products on the treated samples indicated additional cut sites on the target strand and the non-target strand respectively.

For electrophoretic mobility shift analysis (EMSA), DNA substrates were labeled using IR800-conjugated primers. First, 5 μM purified Cas12i effectors were pre-incubated with 10 μM crRNA to form RNP in 1x NEBuffer 2 at 37 °C for 30 min. In vitro binding reactions were performed at 37 °C for 1 h, containing 20 nM labeled dsDNA substrates and serial diluted RNP (500 nM to 37.5 nM) in 1x binding buffer (50 mM NaCl, 10 mM Tris-HCl, 1 mM TCEP, 10% glycerol, 2 mM EDTA, pH 8.0). The reactions were immediately run on a 6% DNA retardation gel (Thermo Fisher #EC6365BOX) for 90 min at 80 V and were imaged on an Odyssey CLx.

**Cell culture and transient transfection.** HEK293T cells were ordered directly from ATCC (Cat: CRL-3216 Lot: 70016639) and expanded and banked upon receipt. No contaminating cell lines (e.g. HeLa) were present in the facility at the time of expansion and banking. Proliferating populations were discarded after passage 20 and replaced with freshly thawed cells from the bank. HEK293T cells were maintained below 90% confluency in D10 media consisting of DMEM plus GlutaMAX and pyruvate (Thermo Fisher #1056-010) supplemented with 10% FBS (Corning #35-010-CV) and 100 U/mL Penicillin-Streptomycin (HyClone #SV30010). Prior to transfection, HEK293T cells were plated in tissue culture treated 96-well plates at 25,000 cells per well in 100 μL of D10. Cells were transfected by Lipofectamine 2000 (Thermo Fisher #11668-019)) 15–18 h after plating. For screening transfections, 182 ng of effector plasmid and 14 ng of purified U6-guide PCR product (1:2 effector:guide molar ratio) were diluted in 10 μL of Opti-MEM media (Thermo Fisher #31985-062) and then mixed with 0.5 μL of Lipofectamine diluted in 9.5 μL of Opti-MEM following the manufacturer's instructions. The DNA plus Lipofectamine solution was then added dropwise to a well of cells. The transfections with plasmid guides contained 126 ng of effector plasmid DNA and 74 ng of guide DNA to maintain the same 1:2 effector:guide molar ratio. Transfected cells were cultured for 72 h. Post-transfection, cells were harvested by removing media and adding 10 μL of TrypLE cell dissociation reagent and incubating at 37 °C for 5 min. Dissociated cells were resuspended in 100 μL D10 media, and cells were pelleted by centrifugation. Supernatant was discarded and pellets were resuspended in 20 μL QuickExtract™ DNA Extraction Solution (Biosearch Technologies/Lucigen #QE9050)) and lysates were subjected to the appropriate thermal profile (15 min at 65 °C, 15 min at 68 °C, 10 min at 98°) to complete cellular and protein degradation.

**Sequencing library preparation.** Target-specific amplification (PCR1) was performed by combining 2 μL of cell lysate with 5 μL NEB Ultra II Q5 Master Mix (New England Biolabs #M0544S), 0.05 μL each of target-specific forward and reverse primer stock at 100 uM, and 3 μL of water. Primers were delivered to wells using an Echo 650 liquid handler. The reaction was mixed well and amplification was performed on a thermocycler (Initial Dentaturation: 60 s at 98 °C. Amplification: 10 s at 98 °C, 15 s at 60 °C, 20 s at 72 °C, cycle count = 25. Final extension: 120 s at 72 °C). Sample indexing (PCR2) was performed by combining 0.5 μL PCR1, 5 μL NEB Ultra II Q5 Master Mix, 0.5 μL each of forward and reverse primer stock at 2uM, and 3 μL of water. Amplification was performed using the same protocol as in PCR1 with 12 cycles of amplification. PCR2 samples were then pooled together and purified using DNA Clean & Concentrator—25 (Zymo Research #D4033). Libraries were sequenced on an Illumina NextSeq 550 system using v2.5 chemistry and a 1 × 150 read length.

**RNP complexing and delivery.** ABR-001 RNP complexation reactions were made by mixing purified ABR-001 (400 μM) with individual crRNAs (1 mM in 250 mM NaCl; sequences in Supplementary Table 5) at a 1:1 (effector:crRNA) volume ratio (2.5:1 crRNA:effector molar ratio). For no guide control, ABR-001 was mixed with 250 mM NaCl at the same volume ratio as the crRNA. SpCas9 RNP complexation reactions were made by mixing SpCas9 (Aldevron; 62 μM) with sgRNA (1 mM in water; sequences in Supplementary Table 5) at a 6.45:1 (effector:sgRNA) volume ratio (2.5:1 sgRNA:effector molar ratio). Complexations were incubated on ice for 30–60 min. prior to T cells or CD34+ HSPCs electroporation. During incubation, T cells or CD34+ HSPCs were harvested and electroporated as described below.

**Ex vivo editing of T cells.** Frozen human Peripheral Blood Mononuclear Cells (PBMCs) (StemCell Technologies #70025) from a donor were revived and counted using an automated cell counter. T cell were isolated from PBMCs using the EasySep Human T Cell Isolation Kit (StemCell Technologies #17951).). Following isolation, a sample was collected and stained for CD3 for flow cytometry analysis of surface expression, to determine T cell purity of the isolated cells. Cell density was adjusted to 1e6 cells/mL and cells were stimulated for 3 days with a cocktail of anti-CD3:CD28 antibodies. Cells were cultured in fresh complete ImmunoCult-XF Cell

Expansion Medium (StemCell Technologies #10981) with 10 ng/mL IL-2 and 2 mM L-Glutamine and supplemented with 25 μL/mL of ImmunoCult Human CD3/CD28 T Cell Activator (StemCell Technologies #10971).

RNP complexation reactions were made as described above. Diluted complexed reactions were dispensed at 2 μL per well into Lonza 16-well nucleocuvette strips. Activated T cell suspensions were collected and counted using an automated cell counter. A sample of cells was collected and stained for CD25 for flow cytometry analysis, to determine activation efficiency. Cell density was adjusted to 1.1e7 cells/mL in P3 buffer (Lonza) and was dispensed at 2e5 cells/reaction (18 μL) into the Nucleocuvette strips. Cell density was adjusted to 1.1e7 cells/mL in P3 buffer (Lonza #VXP-3032) and was dispensed at 2e5 cells/reaction (18 μL) into the Nucleocuvette strips. The strips were electroporated using an electroporation device (program EO-115, Lonza 4D-nucleofector), excluding the unelectroporated conditions. Immediately following electroporation, added 40 μL pre-warmed ImmunoCult-XF+IL-2+L-Glutamine to cells and mixed gently by pipetting. For each technical replicate plate, plated 15 μL (~50,000 cells) of diluted nucleofected cells into pre-warmed 96-well plate with wells containing 200 μL ImmunoCult-XF+IL-2+L-Glutamine. Editing plates were incubated for 7 days at 37 °C with 100 μL media replacement at day 4.

After 7 days, wells were mixed by pipetting and 150 μL from each well was transferred to a fresh 96-well plate for staining. Pellets were spun down @ 400 × g for 10 min. Pellets were then resuspended in 200 μL of PBS. 100 μL of sample was collected and stained with either a fluorescently tagged antibody or LIVE/DEAD stain (Thermo Fisher #L34964) to assess viability.

For genomic DNA extraction, we pelleted the remaining 50 μL of cell suspension by centrifugation, removed the supernatant, and resuspended pellets in 50 μL QuickExtract. The QuickExtract solution was subjected to the appropriate thermal profile (15 min at 65 °C, 15 min at 68 °C, 10 min at 98°) to complete cellular and protein degradation.

**Ex vivo editing of CD34+ HSPCs**. Frozen bone marrow CD34+ cells were thawed and assessed for cell number and viability. CD34+ cells were maintained in culture for about 2 days prior to electroporation. For the in vitro experiments, about 100,000 cells were used per electroporation reaction in a 20 μL format using the P3 primary cell 4D-nucleofector kit S (Lonza #VXP-3032).). For the in vivo experiments, about 1,000,000 cells from a single donor were used per electroporation reaction in a 100 μL format using the P3 primary cell 4D-nucleofector kit L (Lonza #VXP-3024). The cells were mixed with transfection enhancer oligo (~4 μM final concentration) and 10% volume ratio of each RNP complex to a final concentration of 16 μM or 20 μM of ABR-001 RNPs as indicated. The final concentration for the multiplexed guides was 10 μM of each of the RNP complexes. Final concentration of SpCas9 RNPs was 5 μM. Post-electroporation, the cells were mixed with StemSpan SFEM II media and the appropriate supplements and maintained in culture for 3 days. Cell pellets from each test condition were collected after viability testing at 72 h post-electroporation. gDNA extraction and NGS sequencing and analysis were performed as described above.

For erythroid differentiation of the HSPCs, either electroporated HSPCs or untransfected HSPC controls were plated in phase 1 erythroid differentiation medium (EDM1)[24] consisting of base EDM (IMDM supplemented with 5% human serum, 330 g/mL human holo-transferrin, 10 μg/mL recombinant human insulin, 2 IU/mL heparin, and 3 IU/mL EPO), supplemented with 1 M hydrocortisone, 5 ng/mL IL-3, and 100 ng/mL SCF. At day 6 post editing, cells were switched to phase II EDM (base EDM supplemented with 100 ng/mL SCF) for 4 days. From days 10–20 post editing, cells were expanded in phase III EDM3 (base EDM only). At day 20, cells were counted and assessed for intracellular HbF staining. Briefly, the cell samples were fixed, permeabilized and stained with anti-human antibodies for fetal hemoglobin (clone HbF-1; Life Technologies). Remaining cells were collected for gDNA extraction for Indel analysis.

**Colony forming cell (CFC) assay**. To evaluate the progenitor frequency of HSPCs after being electroporated and cultured for 3 days, 1e4 cells were harvested from each condition and added to MethoCult H4435 Enriched (StemCell Technologies) tubes at 250 cells/dish (two MethoCult™ tubes total per test condition). The contents of each tube were then plated into 3 replicate 35 mm dishes. After 15 days of culture at 37 °C and 5% CO₂, the total number of myeloid (CFU-GM), erythroid (BFU-E) and mixed (CFU-GEMM) colonies were enumerated based on morphology. Once counting was complete, photographs were taken of representative colonies from each test condition. The colony number for each test condition was tabulated.

Individual erythroid and myeloid colonies were harvested by plucking single colonies and pipetting them into one well of a 96-well V-bottom plate containing 150 μL of PBS + 2% FBS. Colonies were then pelleted by centrifugation at 739 g for 5 min, supernatant carefully removed and the cell pellets stored at −80 °C. For gDNA extraction, pellets were thawed to room temperature and resuspended in appropriate volume of QuickExtract. Samples were then cycled in a PCR machine at 65 °C for 15 min, 68 °C for 15 min, 98 °C for 10 min. Samples were then frozen at −20 °C. Samples for Next Generation Sequencing (NGS) were prepared and analyzed as described above.

**Flow cytometry and analysis**. PBMCs and T cells were harvested and washed twice with cold FBS stain buffer (BD #554656), fixed in Cytofix fixation buffer (BD 554655) for 15 min at 4 °C, and washed twice with cold FBS stain buffer. Cells were stained with extracellular fluorescently-conjugated antibodies for 15 min, then washed twice with cold FBS stain buffer. For intracellular staining, cells were washed twice with cold 1X permeabilization buffer (BioLegend #421002) and incubated with intracellular fluorescently-conjugated antibodies for 15 min at room temperature, then washed twice with cold FBS stain buffer. Prior to flow cytometry analysis, stained cells were washed twice with cold 1X permeabilization buffer and resuspended in cold FBS stain buffer. A list of antibodies used in this study can be found in Supplementary Table 6.

For mouse peripheral blood and bone marrow cells, cells were collected post-transplantation and washed twice with FACS buffer (PBS + 2% FBS + 1 mM EDTA). Cells were resuspended in Mouse or Human Fc Blocking solution (BD) and incubated at room temperature for 10 min. Cells were incubated with fluorescently-conjugated antibodies for 30 min at 4 °C, then washed twice with FBS buffer. Cells were analyzed using the CytoFLEX (Beckman Coulter).

For viability staining, diluted 7-AAD was added to each well and incubated for 10 min at room temperature. Flow cytometry was performed using a CytoFLEX S instrument (Beckman Coulter), and data were analyzed using FlowJo v10.8.1 (BD). A representative gating example can be found in Supplementary Fig. 21.

**In vivo engraftment of ex vivo edited CD34+ HSPCs**. This work was covered under an ethics protocol reviewed and approved by the Institutional Animal Care Committee at the University of British Columbia under protocol #A18-0276. During the study the care, housing and use of animals was performed in accordance with the Canadian Council on Animal Care Guidelines. Edited and control cells were resuspended at 1,000,000 cells/mL and prepared in PBS + 2% FBS for each of the conditions. 200 μL of cell suspension was then injected into the tail veins of eight groups of sub-lethally irradiated 6–8 week old NOD.Cg-Prkdc^scid Il2^rgtm1Wjl/SzJ (NSG) mice (4–5 mice /group) at a density of 200,000 cells/mouse. In addition 7 days prior to irradiation, and for the duration of the study, acidified water containing antibiotic was provided to the mice. At 8 and 16 weeks after injection, mice were sacrificed and the peripheral blood (PB) and bone marrow (BM) from each individual mouse was harvested and the engraftment of human cells was assessed by flow cytometry with both anti-human and anti-mouse CD45 antibodies. Flow cytometry data were analyzed using FlowJo v10.8.1 (BD). A representative gating example can be found in Supplementary Fig. 22. A portion of the cells were assessed for intracellular fetal hemoglobin staining as described above. The remaining cells were assessed by flow cytometry to determine the presence of human myeloid (CD15/66b), human lymphoid (CD19/20) and human erythroid (CD71) cell populations.

**Delivery of ABR-001 via AAV vector**. Prior to transduction, HEK293T cells were plated at 25,000 cells per well in 100 μL of D10 media into wells of a 96-well plate and grown at 37 °C for 18–24 h to 70–90% confluence. AAV master mixes were prepared by diluting AAV preps in 25 °C D10 media to a concentration double that required for transduction. Cells were transduced by removing 50 μL of media and replacing dropwise with 50 μL AAV particles diluted in D10 to produce a series of twofold dilutions ranging from a AAV GC:Cell ratio of 512000:1 down to 4000:1. Negative controls were prepared by replacing the media of control wells with D10 media only. Post-transduction, cells were harvested and sequencing libraries were prepared as described above.

**Indel pattern analysis**. Indel rates and patterns were analyzed from targeted deep sequencing data. The analysis pipeline sampled up to 50,000 reads and used a kmer-scanning algorithm to calculate the edit operations (match, mismatch, insertion, deletion) between each read and the amplicon reference sequence. The indel rate was calculated as the number of reads containing an insertion or deletion divided by the total number of reads analyzed. Indel patterns were analyzed by aggregating all indel reads for a sample, binning the reads by indel length (size) and calculating the fraction of indel reads at each indel length (size).

**TTISS off-target screen**. The TTISS off-target screen was conducted as previously described[19] with the following modifications. 375 μL 100 ng/μL transposon DNA was mixed with 375 μL glycerol and 750μL EZ-Tn5 Transposase (Lucigen #TNP92110). The solutions were then mixed by vortexing for 5 s and incubated at room temperature for 30 min to form the loaded Tn5 transposome. Transposome was stored at −20 °C until ready for use.

Prior to transfection, HEK293T cells were plated in 24-well tissue culture-treated plates at a density of 125,000 cells per well in 500μL D10 media. Cells were transfected approximately 15 h after plating using GeneJuice transfection reagent (Millipore Sigma #70967) For each well to be transfected, 500 ng donor oligo at ~1.25 ng/μL, 375 ng ABR-001 effector plasmid at 1000 ng/μL, 125 ng guide plasmid at 100 ng/μL were added to Opti-MEM media to a final volume of 125μL. In a separate vessel, 122.5μL Opti-MEM media was mixed with 2.5μL GeneJuice transfection reagent, and this solution was incubated at room temperature for 5 min. Following the incubation, the DNA + Opti-MEM and GeneJuice+Opti-MEM solutions were combined and incubated at room temperature for 5–15 min.

After incubation, the combined solution was added dropwise to a single well of a 24-well plate. Transfected cells were incubated for approximately 72 h.

Cells were then dissociated from the plate by removing media, washing once with 200 µL PBS (Thermo Fisher #10010023), adding 50 µL of TrypLE dissociation reagent (Thermo Fisher #12604013), and incubating at 37 °C for 5 min. Cells were then resuspended by adding 150 µL of D10 media and mixing well. Resuspended cells were then transferred to a 96-well PCR plate and spun down at 400 g for 10 min. The supernatant was removed and cell pellets were stored at −20 °C until DNA extraction.

DNA was extracted as described[19] and gDNA was purified using the Zymo gDNA clean and concentrator-5 kit (Zymo Research #D4067) following the manufacturer's instructions and eluting in 35 µL of 10 mM Tris-HCl. gDNA was visualized on a gel and quantified by the Qubit high-sensitivity dsDNA kit (Thermo Fisher #Q38251) following the manufacturer's instructions. gDNA extracts were then normalized to 50 ng/µL in 10 mM Tris-HCl.

Genomic DNA was tagmented by first preparing a solution 24 µL transposome, 6 µL EZ-Tn5 10X Reaction Buffer, 24 µL purified and diluted genomic DNA, and 6 µL of water. This solution was mixed well and incubated at 37 °C for 2 h. Following the incubation, 6 µL of Ez-Tn5 10X stop solution was mixed into the reaction. The reaction + stop solution mixture was then heated at 70 °C for 10 min. The stopped reaction was then purified using the Zymo gDNA clean and concentrator-5 kit by following the manufacturer's instructions and eluting in 50 µL 10 mM Tris-HCl.

**TTISS Analysis**. TTISS data was analyzed using a bioinformatics pipeline to identify on and off-target sites. The raw sequencing data was first demultiplexed and converted to paired-end fastq files using Illumina's bcl2fastq2 v2-20.0.422 software. Reads that did not begin with the double-stranded oligodeoxynucleotide (dsODN) primer sequence or had low sequencing quality were filtered out, and the remaining reads were truncated to 25 bp of genomic DNA in the forward direction and 15 bp of genomic DNA in the reverse direction. The truncated reads were then aligned to the GRCh38 human reference genome, allowing for a maximum fragment length of 1000 bp. The alignments were filtered to only include read pairs that mapped properly, and PCR duplicate reads were marked. All 100 bp genomic windows with at least two unique aligned reads were identified as possible integration sites, and for each window a cut site was predicted based on the frequency and positional distribution of its aligned reads. Each window was then searched for any sequences within an edit distance of 6 from the guide sequence(s) used for the sample. Windows without any putative on/off-target sequences were removed. Windows where the on/off-target sequence's cut site was over 8 bp from the predicted cut site or where the predicted cut site was not preceded by an NTTN PAM were also removed. Windows that appear in negative control samples were removed to avoid common break sites, mispriming, or other sequencing artifacts. The windows passing these filters were then output as potential on/off-target sites.

**Reporting summary**. Further information on research design is available in the Nature Research Reporting Summary linked to this article.

## Data availability
Data that support the results are available within the article and the Supplementary Information. Source data are provided with this paper. Flow cytometry data has been deposited on FLOWRepository.org under Repository IDs: FR-FCM-Z57L (Fig. 3b, c, Supplementary Fig. 14a–d), FR-FCM-Z57N (Fig. 3e, f), FR-FCM-Z57J (Fig. 3k), FR-FCM-Z57X (Supplementary Fig. 16b (8wk)), FR-FCM-Z57P (Supplementary Fig. 16b (16wk)), FR-FCM-Z57V (Supplementary Fig. 16c (8wk)), FR-FCM-Z57U (Supplementary Fig. 16d (16wk), e-f). All DNA sequencing data is available on the NCBI Sequence Read Archive (SRA) under the Bioproject ID PRJNA829411

## Code availability
Custom code was used for indel analysis though analysis with CRISPresso will yield comparable results. Custom code was used for TTISS analysis, though analysis with BrowserGenome.org will yield comparable results. Code will be made available upon reasonable request.

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

## Acknowledgements

The authors would like to thank Kelly Morgan, John Murphy, Greg Hoffman, Craig Mak, and Devyn Smith for critical reading and comments on the manuscript. We thank Amy Biasella, Austin Jones and Greg Hoffman for helping analyzing in vitro screen data, Lauren Killingsworth and Erin Doherty for contributing to biochemical experiments, and entire Arbor Biotechnologies team for support.

## Author contributions

S.C. and T.D. conceived and designed the project with input from P.H., W.X.Y., and D.A.S. C.M. established in vitro fluorescent reporter assay and performed in vitro screen experiments. G.Z.M., A.J.G., W.X.Y., C.M., W.L., D.C. performed mammalian cell screen experiments. A.J.G., G.Z.M., and Q.W. conducted off-target experiments, prediction and analysis. J.R.H. and S.Se. performed ex vivo primary T cell and CD34+HSPC experiments. A.J.G and M.B. performed AAV experiments. A.O. and P.H. performed biochemical and cut site characterization experiments. E.K.S. J.M.C., S.So., and B.H. performed protein purification and RNP production. E.K.S. conducted EMSA experiments. B.H. conducted structural analyses. P.Y. and H.Z. conducted sample preparation and deep sequencing experiments. L.E.A., R.Z., D.R.C., and D.A.S conducted characterization and phylogenic analysis for Cas12i systems. Q.W., N.J., R.Z., and D.A.S. analyzed all deep sequencing and indel data. S.C. and T.D. wrote the manuscript with input from all authors. S.C. and T.D. contributed equally to this work.

## Competing interests

C.M., A.J.G., G.Z.M., J.R.H., S.S., E.K.S., M.B., A.O., Q.W., N.J., P.Y., H.Z., L.E.A., R.Z., D.R.C., and T.D. are current employees and shareholders of Arbor Biotechnologies (Arbor). W.X.Y. and D.A.S are co-founders and shareholders of Arbor. P.H., J.M.C., W.L., D.C., B.H., S.So., and S.C. are former Arbor employees and were employed when this work was conducted. Some of the content in the manuscript has been included in a patent application published as WO 2021202800 on October 7, 2021. S.C., B.H., Q.W. N.J., R.Z., J.M.C, T.D., J.R.H., A.J.G., C.M., D.A.S., and D.C. are listed as inventors on WO 2021202800.
