## [Peer Review File · Nature Communications]

Reviewers' Comments:

Reviewer #1:

Remarks to the Author:

In this article, the authors engineered Cas12i2 through introducing mutations at critical positions. The engineered variant, ABR001, exhibited robust editing efficiency and high specificity both in immortalized and primary cells. In addition, ABR001 showed a high genome editing efficiency when delivered via AAV to 293T cells. Follow are some questions and suggestions.

1. What is the transfection efficiency of ABR001 RNP into human CD34+ cells? The authors didn't show transfecting control when introducing ABR001 RNP into human CD3+ or CD34+ primary cells. It is necessary to exclude the possibility that those electroporated but not transfected cells interfered the results when investigating whether ABR001 RNP transfected cells maintain multilineage reconstitution.
2. Since ABR001 cause lager indel size than SpCas9, which increase the possibility of genome instability, it is necessary to evaluate its safety. Whether ABR001 transfected CD34+ cells cause other changes in phenotype besides HbF expression variation should be taken into consideration.
3. In the discussion, the authors mentioned improving the editing efficiency of Cas12i2 to "a therapeutically relevant level". Here related references should be cited.

Reviewer #2:

Remarks to the Author:

McGaw and co-authors report development of Cas12i2 as a versatile high-efficiency platform for therapeutic genome editing. The authors conducted in vitro high throughput screening and selected an engineered Cas12i2 variant, termed ABR-001, which could improve the initially low gene-editing efficiency of Cas12i2. They demonstrated that ABR-001 exhibits broad genome editing capability in HEK cells, human cell lines, primary T cells, and CD34+ hematopoietic stem and progenitor cells when delivered by different approaches including plasmid, RNP, and AAV vector.

The manuscript is well written and easy to follow, except for some minor errors. The high throughput mutational scanning method is impressive. I believe this paper is of broad interest to the audience of Nature Communications. However, the following points should be addressed prior to publication.

1. The development of Cas12i as another platform for genome editing is appreciated and important. However, its efficiency and specificity may require a lot of additional and future work, especially when compared to other platforms such as Cas9. I suggest the authors double checking statements on specificity such as 'high specificity' and 'the highest editing levels'. If the specificity is higher than Cas9 and other type V systems, it would be helpful to discuss possible mechanistic explanations.
2. The term ABR-001 should be made consistent to avoid confusion. For example, Cas12i2 in Fig.3 probably is ABR-001 instead of the original version of Cas12i2. Check and change ABR001 to ABR-001 in the second Supplementary Fig. 13. There are two Supplementary Fig. 13.
3. Indicate DNA substrate and cleavage products in Supplementary Fig.7. How many cleavage product bands are there?
4. Supplementary Fig. 16c was not cited in the text.
5. In Supplementary Fig. 16c, the molecular weights of Cas12i2 variants are ~115 kDa as shown in the SDS-PAGE. However, in Supplementary Figure 17, ABR-001 is around ~150 kDa. Can the authors explain the difference?

6. The authors conducted flow cytometry for cell viability measurement and fluorescence-activated cytometry sorting (FACS), but there are no experimental details in the Methods section.

7. Legends for Fig. 3e and 3f may need to be swapped.

8. The FACS histogram in Fig. 3k indicated the percentage of protein expression. To be consistent, the authors could indicate the percentage of each peak in Fig. 3c and 3f. In addition, the indel efficacy of 5 μ M treatment is comparable to that of 10 μ M; however, protein expression levels are significantly different under the conditions. Can the authors explain?

9. The fraction of indels raised by ABR-001 are mostly large deletions of 5-20 nts, in comparison to small deletions by SpCas9. Are the variable deletion lengths raised by ABR-001 related to the wide distribution of cleavage sites as shown in Supplementary Fig. 11?

10. It seems that the ratio between the disruption rate and the raw indel rate (target 2 in Figs. 3h and 3i) reduced over time. Can the authors explain why?

11. In the EMSA assay shown in Supplementary Fig. 14, the intensity of target DNA in the 'Apo' lane is much lower than in the 'Ref' lane. Is this caused by loading errors?

12. Fig. 4c shows indel rate of 80%, which was noted as the highest frequency among type V nucleases. In a recent paper (PMID: 34475560), the indel frequency by Cas12f was shown over 80% as well. The authors should check this paper and discuss.

13. When discussing the recently published Cas12i structures, the authors could also cite the recently published the cryo-EM structure of Cas12i (PMID: 32895556).

14. The authors stated that "ABR-001 delivered via AAV exhibited the highest editing levels demonstrated by a type V nuclease to date". Have they compared ABR-001 with all other type V nucleases, including Cas12b, Cas12e (CasX) and Cas12f?

Reviewers' Comments to the Authors:

Reviewer #1 (Remarks to the Author):

In this article, the authors engineered Cas12i2 through introducing mutations at critical positions. The engineered variant, ABR001, exhibited robust editing efficiency and high specificity both in immortalized and primary cells. In addition, ABR001 showed a high genome editing efficiency when delivered via AAV to 293T cells. Follow are some questions and suggestions.

1. What is the transfection efficiency of ABR001 RNP into human CD34+ cells? The authors didn't show transfecting control when introducing ABR001 RNP into human CD3+ or CD34+ primary cells. It is necessary to exclude the possibility that those electroporated but not transfected cells interfered the results when investigating whether ABR001 RNP transfected cells maintain multilineage reconstitution.

We thank the reviewer for pointing out the possibility that electroporated (but not edited) cells may have interfered with the results. To address this concern, we performed *in vitro* colony forming cell (CFC) assays on CD34+ HSPCs electroporated with ABR-001 RNP, as well as "effector only" controls (i.e. cells electroporated with ABR-001 nuclease but no guide) (line 188-196 and New Supplementary Fig 15). Erythroid (BFU-E), myeloid (CFU-GM), and mixed (CFU-GEMM) colony counts were comparable across RNP and "effector only" control samples, suggesting that ABR-001 RNP transfected cells maintain multilineage differentiation potential. While transfection efficiency was not directly measured, we observed >70% indel activity in multiple RNP samples at 72 hours post transfection, indicating high transfection efficiency. Furthermore, indels were maintained over the course of the CFC differentiation assay (15 days). This data is further supported by our *in vivo* lineage analysis from mouse peripheral blood and bone marrow, which demonstrated comparable erythroid, lymphoid, and myeloid populations at 16 weeks post transplantation across treated and control conditions (Supplementary Fig 16). However, since the animal study did not include the transfecting "effector only" control, we removed the reference to *in vivo* multilineage reconstitution in the main text (225-228).

2. Since ABR001 cause larger indel size than SpCas9, which increase the possibility of genome instability, it is necessary to evaluate its safety. Whether ABR001 transfected CD34+ cells cause other changes in phenotype besides HbF expression variation should be taken into consideration.

As these are illustrative experiments of therapeutic potential of our enzyme, we agree that additional safety measures should be assessed. While additional work is required to fully evaluate ABR-001 effects, our initial assays and characterizations of ABR-001 edited CD34+ HSPCs suggest that cellular phenotype is maintained. For example, we used *in vitro* CFC assays (New Supplementary Fig 15) and *in vivo* engraftment studies (Supplementary Fig 16) to broadly assess the function and phenotype of the ABR-001-edited CD34+ HSPCs. With these experiments, we've

shown successful editing of a human stem cell without impacting the viability, ability of the cell to engraft, or the ability of the cell to differentiate and express the biologically meaningful endpoints. In addition to characterizing the indel size, we extensively characterized off-target nuclease activity in cell lines (Figure 2). However, we agree with the reviewer that additional functional experiments and molecular characterization would be needed to advance ABR001-treated CD34+ HSPCs to the clinic, but such experiments are outside the scope of this paper.

3. In the discussion, the authors mentioned improving the editing efficiency of Cas12i2 to “a therapeutically relevant level”. Here related references should be cited.

We have included references to seminal papers describing edited T cell therapy and edited CD34+ cell therapy (line 262).

We have also modified broad language regarding therapeutic potential throughout the paper given that *ex vivo* editing of T cells and CD34+ HSPCs only represent a few of the many potential therapeutic applications of gene editing technology (line 78-79).

Reviewer #2 (Remarks to the Author):

McGaw and co-authors report development of Cas12i2 as a versatile high-efficiency platform for therapeutic genome editing. The authors conducted in vitro high throughput screening and selected an engineered Cas12i2 variant, termed ABR-001, which could improve the initially low gene-editing efficiency of Cas12i2. They demonstrated that ABR-001 exhibits broad genome editing capability in HEK cells, human cell lines, primary T cells, and CD34+ hematopoietic stem and progenitor cells when delivered by different approaches including plasmid, RNP, and AAV vector.

The manuscript is well written and easy to follow, except for some minor errors. The high throughput mutational scanning method is impressive. I believe this paper is of broad interest to the audience of Nature Communications. However, the following points should be addressed prior to publication.

We appreciate the reviewer's encouraging comments and have adjusted the errors that were pointed out as detailed below.

1. The development of Cas12i as another platform for genome editing is appreciated and important. However, its efficiency and specificity may require a lot of additional and future work, especially when compared to other platforms such as Cas9. I suggest the authors double check statements on specificity such as ‘high specificity’ and ‘the highest editing levels’. If the specificity is higher than Cas9 and other type V systems, it would be helpful to discuss possible mechanistic explanations.

We agree with the reviewer that additional future work may be required to comprehensively compare ABR-001 to other editing systems. Given this feedback, we have limited broad

comparative statements of ABR-001 to other systems and focused the interpretation to the data derived in the head-to-head comparisons (line 111-113, 137-138, line 133, line 289-290). Additionally, we have removed comments comparing ABR-001 AAV to other typeV systems since we did not directly compare ABR-001 to other TypeV AAV systems in this study (line 245-250).

To strengthen our mechanistic discussion, we have added in additional references to support possible mechanisms underlying the improvement of ABR-001 with our engineering strategy (line 269-271), and we have made it clear that the exact mechanism for improvement is unknown but may be elucidated as additional structural and rational engineering data is derived (line 278-281).

2. The term ABR-001 should be made consistent to avoid confusion. For example, Cas12i2 in Fig.3 probably is ABR-001 instead of the original version of Cas12i2. Check and change ABR001 to ABR-001 in the second Supplementary Fig. 13. There are two Supplementary Fig. 13.

Thank you for pointing out the labeling inconsistency as well as the Supplementary numbering. We have updated all instances of "ABR001" to "ABR-001" and changed "Cas12i2" to "ABR-001" where appropriate. Furthermore, we have updated the numbering of the Supplementary Figures as well as the associated references in the text.

3. Indicate DNA substrate and cleavage products in Supplementary Fig.7. How many cleavage product bands are there?

Thank you for the suggestion. We have added arrows to indicate the 4 cleavage product bands in the assay we performed to Supplementary Figure 7.

4. Supplementary Fig. 16c was not cited in the text.

Thank you for pointing this out. We have added reference to Supplementary Fig 16c (now 19c) to the main text (line 312).

5. In Supplementary Fig. 16c, the molecular weights of Cas12i2 variants are ~115 kDa as shown in the SDS-PAGE. However, in Supplementary Figure 17, ABR-001 is around ~150 kDa. Can the authors explain the difference?

Thank you for pointing this out. The molecular weight standards in the gels from Supplementary Figure 16c (now 19c) and Supplementary Figure 17 (now 20) are in fact the same ladder; however, the standards were mislabeled in Supp. Figure 17 (now 20) and has since been corrected. As expected, the cell-free expressed protein in Supp. Fig. 16c (now 19c) and the purified protein in Supp. Fig. 17 (now 20) are both at ~115kDa.

6. The authors conducted flow cytometry for cell viability measurement and fluorescence-activated cytometry sorting (FACS), but there are no experimental details in the Methods section.

We apologize for this oversight and thank the reviewer for the suggestion. We have included a new Methods section detailing the flow cytometry analysis performed in this manuscript (line 597-617). However, since fluorescence-activated cytometry sortings (FACS) was not performed in this study, we have not included it in the Methods section.

We also have added Supplementary Table 6 with all of the antibodies used in this study.

7. Legends for Fig. 3e and 3f may need to be swapped.

The legends for the two referenced subplots were indeed out of order and have been swapped to correct the issue. Thank you for bringing our attention to this.

8. The FACS histogram in Fig. 3k indicated the percentage of protein expression. To be consistent, the authors could indicate the percentage of each peak in Fig. 3c and 3f. In addition, the indel efficacy of 5 μ M treatment is comparable to that of 10 μ M; however, protein expression levels are significantly different under the conditions. Can the authors explain?

We thank the reviewer for the suggestion. We have modified Fig. 3c and 3f to indicate the percentage of each peak.

Regarding the correlation between B2M indel efficacy and protein expression of 5 μ M vs. 10 μ M treatment, the plot shown in Fig. 3c is representative of one technical replicate, while the indel data in Fig. 3a is representative of the average of three donors with multiple technical replicates each.

While B2M protein expression variability was observed across donors and replicates at the 5 μ M RNP concentration, we observed consistently high knockdown of B2M at the saturating doses of 16 μ M and 10 μ M concentrations. To account for the variability seen across technical replicates, we have added an additional supplementary figure (Supplementary Figure 14) that shows all technical replicates for flow. In addition, we chose a more representative technical replicate to feature in the main text as Fig. 3c.

9. The fraction of indels raised by ABR-001 are mostly large deletions of 5-20 nts, in comparison to small deletions by SpCas9. Are the variable deletion lengths raised by ABR-001 related to the wide distribution of cleavage sites as shown in Supplementary Fig. 11?

We agree with the reviewer's observation that Cas9 generates a more uniform cut pattern. We hypothesize that the variable cut pattern of ABR-001 *in vitro* may be related to the broad distribution of edits in mammalian cells.

10. It seems that the ratio between the disruption rate and the raw indel rate (target 2 in Figs. 3h and 3i) reduced over time. Can the authors explain why?

The reviewer's observation that the relative GATA disruption decreased over time is correct for this experiment. We hypothesize the reason for the discrepancy between input (3 days) and output (20 days) is related to the heterogeneity of the starting population of HSPCs and a discrepancy in delivery/editing to stem and progenitor cells as compared to non-progenitor cells. Specifically, disruption appears to be cell type dependent with the GATA disruption in the stem cell population being different from the GATA disruption of the total heterogeneous starting population. We hypothesize that the differentiated day 20 GATA and raw indel rates are more representative of the disruption rates of the true stem cells within the population.

11. In the EMSA assay shown in Supplementary Fig. 14, the intensity of target DNA in the 'Apo' lane is much lower than in the 'Ref' lane. Is this caused by loading errors?

It is our hypothesis that the lower intensity in the apo lane is most likely due to non-specific binding of the Effector protein to the DNA of interest that leads to aggregation. We have found that the high concentration of apo protein in the "apo" sample (500nM) consistently leads to a strong band that fails to migrate out of the well in EMSA experiments. We hypothesize that the effector protein binding non-specifically to the target DNA leads to aggregation and a lack of migration in the gel, and therefore the un-bound target DNA band is fainter while a resulting aggregation band becomes present in the well.

Supplementary Fig 14 (now 16) has been modified to include the uncropped version of the gel to show the intensity of bands that remain in the well after running, which we suggest indicates aggregated, non-specifically target-bound protein.

12. Fig. 4c shows indel rate of 80%, which was noted as the highest frequency among type V nucleases. In a recent paper (PMID: 34475560), the indel frequency by Cas12f was shown over 80% as well. The authors should check this paper and discuss.

Thank you for calling our attention to the referenced paper by Kim et al. After reviewing the paper and discussing, we have decided to amend our statement regarding ABR-001 editing compared to other type V nucleases since we did not directly compare ABR-001 to other TypeV AAV systems in this study (line 245-250).

13. When discussing the recently published Cas12i structures, the authors could also cite the recently published the cryo-EM structure of Cas12i (PMID: 32895556).

Thank you for calling our attention to the referenced paper by Zhang et al. We have included reference to the paper in the main text in the discussion of modeling the ABR-001 mutations in recent structures (lines 279, 281).

14. The authors stated that "ABR-001 delivered via AAV exhibited the highest editing levels demonstrated by a type V nuclease to date". Have they compared ABR-001 with all other type V nucleases, including Cas12b, Cas12e (CasX) and Cas12f?

This statement was based on available literature data for other type V nucleases, however, since we did not directly compare ABR-001 to other Type V systems in AAV in this study, we have removed this comparative statements and focused the interpretation derived in the head-to-head comparison of ABR-001 to SaCas9 (line 245-250, line 289-290).

Reviewers' Comments:

Reviewer #1:

Remarks to the Author:

The authors have performed experiments which set a negative "effector only" control to exclude the possible interference from not-edited cells. In general, the authors have answered previous concerns, yet there are questions about the CFC assays:

1. The CFC showed no significant difference in the total number as well as proportion of each colony (BFU-E, CFU-GM and CFU-GEMM), which suggests ABR-001 RNP transfected cells maintain multilineage differentiation potential. In the article the guides were designed to generate indels with BCL11A enhancer region to disrupt the GATA motif. Why no variation in erythroid phenotype was observed with a disrupted GATA motif and elevated HbF expression? It is supposed that the number of BFU-E may increase.

2. In this article, the authors performed transplantation and CFC experiments to evaluate ability of engraftment and differentiation potential of edited HSPCs, but not self-renewal, which needs more data from experiments such as serial transplantations or LTC-IC. I agree that the experiments regarding safety evaluation are out of the scope of this paper. Since there is no further exploration to the clinic, it is suggested to modify the terms such as "gene therapy" (line 294-295).

3. Moreover, the sentence "ABR-001 as a highly active and specific genome editing nuclease" (line 150) may need to be modified since the word "highly" has been deleted in the subtitle.

Reviewer #2:

Remarks to the Author:

The authors adequately addressed the concerns raised in last round. The authors may need to check more carefully on spelling and grammar. For example, '0.05uL' should be spelled as '0.05 μ L'.

Reviewer #1 (Remarks to the Author):

The authors have performed experiments which set a negative “effector only” control to exclude the possible interference from not-edited cells. In general, the authors have answered previous concerns, yet there are questions about the CFC assays:

1. The CFC showed no significant difference in the total number as well as proportion of each colony (BFU-E, CFU-GM and CFU-GEMM), which suggests ABR-001 RNP transfected cells maintain multilineage differentiation potential. In the article the guides were designed to generate indels with BCL11A enhancer region to disrupt the GATA motif. Why no variation in erythroid phenotype was observed with a disrupted GATA motif and elevated HbF expression? It is supposed that the number of BFU-E may increase.

We thank the reviewer for pointing out this nuance. While variation in erythroid phenotype has been reported when targeting the coding region of the *BCL11A* gene (Chang et.al, *Mol Ther Methods Clin Dev*, 2017), targeting the *BCL11A* enhancer, as we have done in this work, has been shown to have no impact on multilineage differentiation potential (Psatha et. al, *Mol Ther Methods Clin Dev*, 2018). Hence, our observation that targeting the *BCL11A* enhancer with ABR-001 or SpCas9 has no impact on the lineage or differentiation potential is in line with published findings. Importantly, we also show that targeting the *BCL11A* enhancer with ABR-001 results in the therapeutically meaningful outcome of increased HbF expression.

2. In this article, the authors performed transplantation and CFC experiments to evaluate ability of engraftment and differentiation potential of edited HSPCs, but not self-renewal, which needs more data from experiments such as serial transplantations or LTC-IC. I agree that the experiments regarding safety evaluation are out of the scope of this paper. Since there is no further exploration to the clinic, it is suggested to modify the terms such as “gene therapy” (line 294-295).

We agree with the reviewer that more data is needed to apply ABR-001 as a genetic medicine. We have taken care to qualify statements around therapeutic utility and gene therapy as potential rather than claiming therapeutic efficacy at this point in time.

3. Moreover, the sentence “ABR-001 as a highly active and specific genome editing nuclease” (line 150) may need to be modified since the word “highly” has been deleted in the subtitle.

We have removed the non-quantitative descriptor and adjusted this sentence to read: “...ABR-001 as an active and specific genome editing nuclease.”

Reviewer #2 (Remarks to the Author):

The authors adequately addressed the concerns raised in last round. The authors may need to check more carefully on spelling and grammar. For example, ‘0.05uL’ should be spelled as ‘0.05 μ L’.

We thank the reviewer for pointing this out and have reexamined the manuscript for grammar, spelling, and formatting errors including the use of μ L as pointed out above.